# An extended and generalized framework for the calculation of metabolic intervention strategies based on minimal cut sets

**Philipp Schneider**, **Axel von Kamp**, **Steffen Klamt***

Max Planck Institute for Dynamics of Complex Technical Systems, Magdeburg, Germany

* klamt@mpi-magdeburg.mpg.de

**Data Availability Statement:** Data are contained within the manuscript, the Supporting Information files and on the GitHub repository https://github.com/ARB-Lab/MCS_extended.

## Abstract

The concept of minimal cut sets (MCS) provides a flexible framework for analyzing properties of metabolic networks and for computing metabolic intervention strategies. In particular, it has been used to support the targeted design of microbial strains for bio-based production processes. Herein we present a number of major extensions that generalize the existing MCS approach and broaden its scope for applications in metabolic engineering. We first introduce a modified approach to integrate gene-protein-reaction associations (GPR) in the metabolic network structure for the computation of gene-based intervention strategies. In particular, we present a set of novel compression rules for GPR associations, which effectively speedup the computation of gene-based MCS by a factor of up to one order of magnitude. These rules are not specific for MCS and as well applicable to other computational strain design methods. Second, we enhance the MCS framework by allowing the definition of multiple target (undesired) and multiple protected (desired) regions. This enables precise tailoring of the metabolic solution space of the designed strain with unlimited flexibility. Together with further generalizations such as individual cost factors for each intervention, direct combinations of reaction/gene deletions and additions as well as the possibility to search for substrate co-feeding strategies, the scope of the MCS framework could be broadly extended. We demonstrate the applicability and performance benefits of the described developments by computing (gene-based) *Escherichia coli* strain designs for the bio-based production of 2,3-butanediol, a chemical, that has recently received much attention in the field of metabolic engineering. With our extended framework, we could identify promising strain designs that were formerly unpredictable, including those based on substrate co-feeding.

## Author summary

The targeted modification of metabolic networks, e.g. for designing microbial cell factories or to combat cancer cells, is often supported by computational methods. The framework of Minimal Cut Sets (MCS) uses a constraint-based approach to determine a minimum set of reaction deletions in a metabolic network that enforce desired phenotypes according to user-defined specifications. In this work we generalize the MCS

**Funding:** PS and SK received funding from the European Research Council (Grant 721176). The funders had no role in study design, data collection and analysis, decision to publish, or preparation of the manuscript.

approach by introducing several new features making it suitable for a broader range of applications. Among other extensions, the new features support (1) the combination of multiple strain design specifications at once and thus more precise metabolic network tailoring, (2) the optional addition of alternative substrates or of heterologous reactions in combination with reaction deletions, and (3) an improved direct computation of gene-based intervention strategies also exploiting new compression rules for gene-reaction-enzyme relationships. We use the example of designing *E. coli* strains with different specifications for growth-coupled production of 2,3-butanediol to demonstrate the functional and performance benefits of our methodological enhancements.

This is a *PLOS Computational Biology* Methods paper.

## Introduction

Metabolic engineering of cell factories has become an essential technology for developing bio-based production processes as sustainable alternatives to conventional petrochemical syntheses [1]. Various computational methods and strain optimization approaches have been developed to support the analysis and rational design of metabolic networks in microbial strains, most of which belong to the class of constraint-based modeling approaches (for reviews see [2–4]). OptKnock [5] was one of the first algorithms for metabolic design and introduced the principle of growth-coupled product synthesis. It identifies suitable knockouts in the metabolic network that ensure product synthesis under assumption of optimal microbial growth. OptKnock is based on a bi-level optimization problem, which can be converted to a single-level Mixed Integer Linear Program (MILP). It stimulated the development of a number of related methods such as RobustKnock [6], OptORF [7], OptStrain [8], or MOMAKnock [9], to name only a few. Common to all these methods (sometimes called "biased" strain design methods [3]) is the optimization of product synthesis around a certain operation point, typically at maximal growth rate. Since solving the respective MILP in a genome-scale network can be a delicate problem, simulation-based metaheuristics like evolutionary algorithms may represent good alternatives [10,11]. While they cannot guarantee the attainment of global optima, they allow handling of more complex objective functions and may perform better on bi-level problems that are hard to transform entirely into MILPs.

Another class of computational strain optimization methods is based on the framework of Minimal Cut Sets (MCS) [12–15]. Initially, the concept of MCS was introduced to enumerate all combinations of knockouts that eliminate certain functionalities in the network, e.g., growth (synthetic lethalities) or synthesis of undesired products [12,15]. The potential of using MCS for metabolic network design was pointed out later. When designing a production host, one would target steady-state flux vectors that synthesize undesired products or that lead to poor product yields. However, the deletions needed to eliminate the targeted (undesired) phenotypes can interfere with parts of the metabolic network that are essential for the desired phenotypes of the strain (growth with high product yields). Therefore, the extended framework of *constrained* MCS introduced the option to demand the conservation of desired behaviors [14]. This was an important step for establishing MCS as a strain design method in metabolic engineering. Examples of successful applications of MCS for designing microbial cell factories with

excellent product yields include the growth-coupled synthesis of itaconic acid [16] by *Escherichia coli* and of a heterologous secondary metabolite (indigoidine) by *Pseudomonas putida* [17]. In the latter case, as many as 14 multiplexed gene knockdowns were implemented as suggested by MCS analysis. A large-scale computational study based on the MCS framework also proved that growth-coupled product synthesis is—at least stoichiometrically—feasible for a broad range of metabolic products thus providing a widely applicable strain design principle [18].

The computation of MCS was originally based on elementary modes (EMs). The user needed to specify a set of undesired (target) EMs and, in the case of constrained MCS, a set of EMs that exhibit the desired behavior (*desired modes*) from which at least some must not be eliminated by the cuts. The search for MCSs is then equivalent to the search for *minimal hitting sets* of these EMs [12,14,19]. The bottleneck of MCS computation with this approach was the inevitable preceding enumeration of EMs, which is still infeasible in genome-scale models due to the large number of EMs [20]. Here, the insight that MCS in the primal network are EMs in a corresponding dual network [21] helped overcoming these limitations. The duality principle was used to develop the MCSEnumerator, an algorithm that computes the shortest MCSs from a specified *target region* and a specified *desired region* instead of target and desired EMs [13,18]. This duality approach enabled for the first time the enumeration of thousands of MCS in genome-scale models and initiated the developments of other variants of duality-based MCS calculations ([22], [23]). cRegMCSs, another extension of the MCS framework, allows the use of regulatory interventions, i.e. up- and down-regulation of fluxes, alongside with reaction knockouts [24]. Yet, even with this extension, the MCS considered only *reaction* knockouts or regulations. To account for genetic interventions, Machado et al. [25] proposed to incorporate gene-protein-reaction (GPR) associations into the metabolic network and to use the MCSEnumerator to compute strain designs on a genetic level. Another technique, the gene MCS (gMCS) framework, maps the relationships between sets of gene deletions and their resulting reaction deletions before the MCS are computed. This approach was designed to find lethal gene knockouts in cancer cells [26,27], however, the feature of conserving desired behaviors was not integrated in this approach.

In this work, we introduce a number of extensions and generalizations for the MCS framework that improve algorithmic performance and broaden the scope of applications, especially in the context of strain design. The main extensions include (1) the specification of multiple target and desired regions; (2) the possibility to consider combinations of reaction deletions and additions; (3) the computation of substrate co-feeding strategies; (4) an improved technique for integrating gene-protein-reaction (GPR) associations into the metabolic models and a number of effective preprocessing and compression steps to reduce the GPR rules and thus to accelerate the computation of gene MCS; and (5) the possibility to associate individual cost factors to each intervention. We demonstrate the relevance and applicability of these developments by computing MCS for realistic examples of genome-scale strain designs of *Escherichia coli* for the growth-coupled production of 2,3-butanediol. In particular, we could identify promising new strain designs that were formerly unpredictable, e.g. based on substrate co-feeding. The example calculations also confirm considerable runtime speedups when using the introduced rules for compressing GPR associations.

## Methods

### Definitions and state of the art

We consider a metabolic network with $m$ metabolites and $n$ reactions whose structure is captured in a stoichiometric matrix $\mathbf{N} \in \mathbb{R}^{m \times n}$. In steady state, where the metabolite

concentrations are constant, the reaction rate vector $\mathbf{r} \in \mathbb{R}^n$ fulfills the equation

$$\mathbf{Nr} = \mathbf{0} \tag{1}$$

Individual reaction rates $r_i$ may be constrained with upper ($ub_i$) and lower ($lb_i$) bounds expressing known irreversibilities ($lb_i \geq 0$) and physiological flux ranges (e.g. substrate uptake or ATP maintenance demand):

$$lb_i \leq r_i \leq ub_i \tag{2}$$

By *Irrev* we denote the indices of all irreversible reactions, i.e. *Irrev* = $\{i \mid lb_i \geq 0\}$, and with *Rev* the indices of all reversible reactions, i.e. *Rev* = $\{i \mid lb_i < 0\}$. Note that the flux bounds (2) can also be expressed as

$$\begin{aligned} r_i &\leq ub_i \\ -r_i &\leq -lb_i \end{aligned} \tag{3}$$

and we will later make use of this representation. In some cases we only need the subset of the flux bound constraints expressing pure irreversibility constraints:

$$r_i \geq 0 \ \forall i \in Irrev \tag{4}$$

In addition to flux bounds, other linear (in)equalities, e.g. from enzyme allocation constraints [28,29] may exist that further constrain the possible space of steady-state flux vectors:

$$\mathbf{Ar} \leq \mathbf{b} \tag{5}$$

The set of solutions $\mathbf{r}$ obeying Eqs (1), (2) and (5) spans the space of feasible steady-state flux vectors, also called the (flux) polyhedron. For the calculation of MCS, one first needs to specify a target region which contains the undesired steady-state fluxes that must all be eliminated by the knockouts of an MCS. The target region (more precisely, the target polyhedron) forms a subset (or sub-polyhedron) of the flux polyhedron and can conveniently be characterized by inequalities posed by a matrix $\mathbf{T} \in \mathbb{R}^{t \times n}$ and a vector $\mathbf{t} \in \mathbb{R}^t$:

$$\mathbf{Tr} \leq \mathbf{t} \tag{6}$$

As a typical example in computational strain design, (6) may contain an inequality that describes (undesired) flux vectors with a low product yield below a given threshold (P: product; S: substrate; $r_p$: product synthesis rate; $r_s$: substrate uptake rate, $Y_{min}^{P/S}$: lower product yield threshold):

$$\frac{r_P}{r_S} \leq Y_{min}^{P/S} \ \Leftrightarrow \ r_P - Y_{min}^{P/S} \cdot r_s \leq 0 \tag{7}$$

The case of the (undesired) operation of one or several (target or objective) reactions as originally introduced in [15,30] can also be expressed by (6). Importantly, the target region is defined by specific target constraints (6) together with the general network constraints (1), (3) and (5). For technical reasons, we directly incorporate the flux bounds (3) (excluding pure irreversibility constraints (4); these will be treated separately) and the inhomogeneous constraints (5) in (6); the target region is then spanned by (1), (4) and (6).

In addition to the target region, a *desired region* can be defined which contains the wanted stationary flux vectors of which at least some must be kept feasible after introducing the interventions of an MCS. As for the target region, the desired region (desired polyhedron) can be

specified by suitable inequalities using an appropriate matrix $\mathbf{D} \in \mathbb{R}^{d \times n}$ and a vector $\mathbf{d} \in \mathbb{R}^d$:

$$\mathbf{Dr} \leq \mathbf{d} \tag{8}$$

Inequalities (8) may, for example, express a demanded minimum product yield

$$\frac{r_P}{r_S} \geq Y_{min}^{P/S} \quad \Leftrightarrow \quad -r_P + Y_{min}^{P/S} \cdot r_s \leq 0 \tag{9}$$

or/and that a reaction rate, e.g. the growth rate $r_{BM}$, can operate above a certain threshold:

$$r_{BM} \geq 0.1 \quad \Leftrightarrow \quad -r_{BM} \leq -0.1 \tag{10}$$

For example, the combination of the target region (7) with the desired region defined by (9) and (10) enforce coupling of growth with product synthesis [18,31], a frequently used strain design principle. Again, the actual desired region is contained in the flux polyhedron and is defined by (8) in combination with the general network constraints (1), (2) and (5). Similar as for the target region, we assume that constraints of type (5) are incorporated in the specification of the desired system (8), but, in contrast to the target system, for reasons shown below, the flux bounds remain here separated. Hence, the desired region is now spanned by (1), (2) and (8). Depending on the application, the desired space can also be absent if the only goal is solely to inhibit certain flux vectors. This is the case, for example, when computing synthetic lethals [13,27].

As has been shown in [21] and [13] together with extensions in [32] and [18], based on the Farkas Lemma and duality theory, a system can be constructed from which the MCS can be computed that, as demanded, block all flux vectors in the target polyhedron and keep some flux vectors of the desired system feasible (Fig 1, first row):

$$\begin{pmatrix} \mathbf{N^T} & \mathbf{I} & \mathbf{T^T} & \mathbf{0} \\ \mathbf{0} & \mathbf{0} & \mathbf{t^T} & \mathbf{0} \\ \mathbf{0} & \mathbf{0} & \mathbf{0} & \mathbf{N} \\ \mathbf{0} & \mathbf{0} & \mathbf{0} & \mathbf{D} \end{pmatrix} \begin{pmatrix} \mathbf{u} \\ \mathbf{v} \\ \mathbf{w} \\ \mathbf{r} \end{pmatrix} \begin{matrix} = \\ \leq \\ = \\ \leq \end{matrix} \begin{pmatrix} \mathbf{0} \\ -c \\ \mathbf{0} \\ \mathbf{d} \end{pmatrix}$$

$$z_{p,i} = 0 \rightarrow v_i \leq 0 \, , z_{p,i} = 1 \rightarrow v_i \geq c$$

$$\forall i \in Rev: \quad z_{n,i} = 0 \rightarrow v_i \geq 0 \, , z_{n,i} = 1 \rightarrow v_i \leq -c \tag{11}$$

$$\forall i \in Irrev: z_{n,i} = 0$$

$$z_{p,i} + z_{n,i} = z_i$$

$$(1 - z_i) \cdot lb_i \leq r_i \leq (1 - z_i) \cdot ub_i$$

$$\mathbf{u} \in \mathbb{R}^m, \quad \mathbf{t} \in \mathbb{R}^t, \quad \mathbf{w} \in \mathbb{R}_{\geq 0}^t, \quad c > 0, \quad \mathbf{v}, \mathbf{r} \in \mathbb{R}^n, \quad z_{p,i}, z_{n,i}, z_i \in \{0,1\}$$

The blocking of the target region is addressed in the first two rows of the matrix in (11) via a duality-based relationship where the stoichiometric matrix $\mathbf{N}$ and the matrix $\mathbf{T}$ for the target region are used with their transposed version (cf. [21]). The associated variables for this dual subsystem are $\mathbf{u}$, $\mathbf{v}$ and $\mathbf{w}$. We note that there are variants in treating the $n \times n$ identity matrix $\mathbf{I}$ and its associated vector $\mathbf{v}$ (for example, reversible and irreversible reactions can be separated

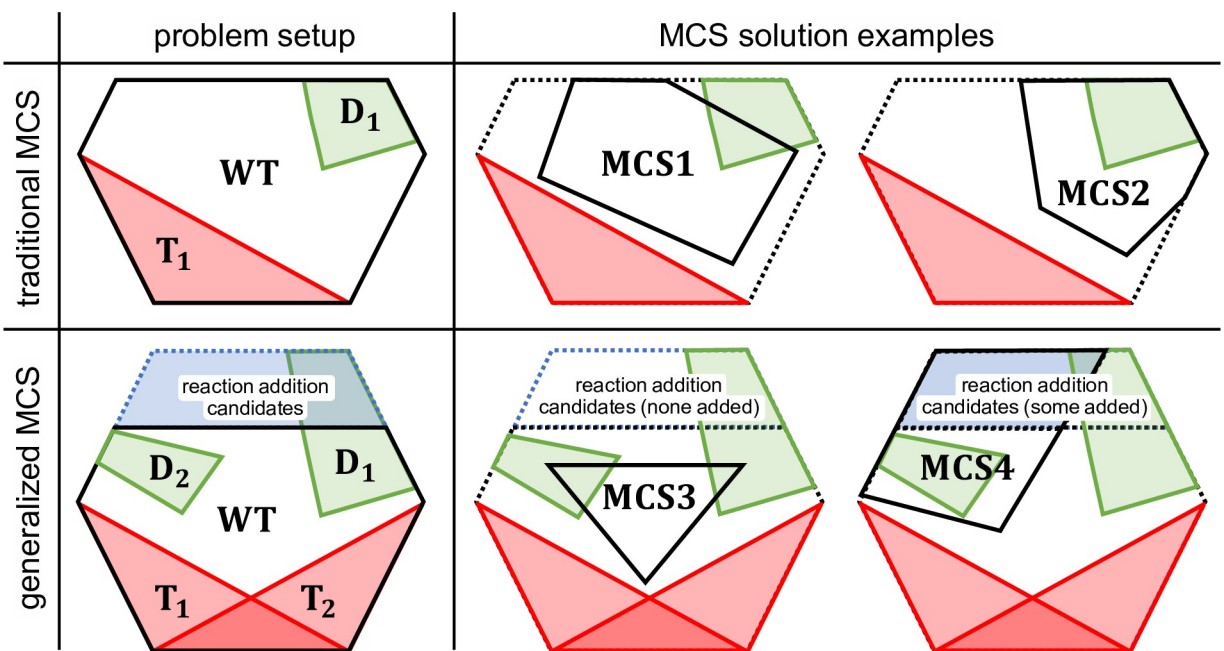

**Fig 1. Exemplary illustration of metabolic solution spaces (solid black lines) of wild type and mutant strains for traditional and extended/ generalized MCS computation.** Red areas: target regions, green areas: desired regions, blue area: flux states accessible through the addition of reactions.

and the reversible be split into a positive and a negative part [18]) but we choose the above version, which is more concise. In contrast to the target region, the requirements for the desired region (third and fourth row in the matrix in Eq (11) as well as the flux bounds for the associated flux variable **r**) are given in their primal representation directly expressing Eqs (1), (2) and (8). The continuous dual (**u**, **v**, **w**) and primal (**r**) variables are interconnected via the binary $z_i$ variables. Each $z_i$ is linked with $v_i$ via two other binary variables ($z_{p,i}$ and $z_{n,i}$). Effectively, cuts are marked by the $z_i$ that are equal to 1, which happens if the associated dual variable $v_i$ of a reversible reaction $i$ is non-zero or if $v_i$ of an irreversible reaction $i$ is positive (hence, negative entries for irreversible reactions in **v** do not count). The smallest MCS can be found by a mixed-integer linear optimization problem (MILP) with an objective function that minimizes the number of cuts:

$$minimize \sum_i z_i \tag{12}$$

As mentioned above, the computed MCS $C$ is given by $C = \{i \mid z_i = 1\}$. Furthermore, multiple MCS solutions (with increasing cardinality) can be found by adding integer cut constraints to the MILP for a previously found solution. If $C_k$ denotes the set of reaction indices that are knocked-out in the $k$-th MCS, then the integer cut constraint reads

$$\sum_{j \ \in C_k} z_j \leq |C_k| - 1 \tag{13}$$

A new MILP optimization based on (11)–(13) is then performed. Furthermore, instead of searching for an MCS with minimal cardinality via the objective function (12), one may also search for an arbitrary feasible solution for (11). This solution would deliver a cut set $C$, but

not necessarily a minimal (irreducible) cut set, i.e. $C$ might contain redundant knockouts. However, by applying, in a second step, objective function (12) to a confined MILP setup, where only the found interventions in $C$ are allowed, a true (support-minimal) MCS will be identified. Still, this MCS is not necessarily an MCS with the fewest number of interventions. Nevertheless, a larger sample of MCS can be quickly generated in this way, which is often faster than always demanding to find the cardinality-minimal MCS in the next iteration.

When particular reactions are not allowed to be knocked-out (e.g., because they are not associated with a targetable gene), their indicator variables can be fixed to $z_i = z_{p,i} = z_{n,i} = 0$, or completely be removed from the system together with $v_i$ and the corresponding columns in the matrix in (11). Furthermore, the number of interventions can be bounded by

$$\sum_i z_i \leq \text{maxCuts}$$

(14)

## Extension 1: Multiple target and desired regions

So far, only one target and one desired region can be considered in (11). This is sufficient for many applications, however, there are cases where multiple target regions or/and multiple desired regions must be defined to properly formulate a strain or network design problem. The idea of multiple desired regions was already discussed in [14]. However, applications with multiple target regions were not considered and, most importantly, the calculation of MCS for multiple desired regions was so far only possible for the classical way of MCS calculation via elementary modes, which, in contrast to the duality-based approach, is not applicable to genome-scale models.

To motivate the use of multiple target regions, we consider an example for the design of a two-stage process. The cells should grow aerobically with high biomass yield in the first phase which requires to block synthesis of acetate as byproduct of overflow metabolism. For the anaerobic production phase we demand a minimum product per substrate yield ($Y_{min}^{P/S}$ e.g., ethanol per glucose) and allowance of some minimum growth rate. Accordingly, we have two target regions that need to be blocked. The first covers flux vectors with production of acetate above a certain minimum value (here set to 1)

$$-r_{Ac} \leq -1$$

which can be written in standard form as (again, $\mathbf{T_1}$ and $\mathbf{t_1}$ also include the constraints (5)):

$$\mathbf{T_1 r} \leq \mathbf{t_1}$$

The second target region comprises anaerobic flux vectors with product per substrate yields below a threshold $Y_{min}^{P/S}$. Importantly, the trivial (zero) flux vector must not belong to the target region, because this flux distribution can never be eliminated through knockouts and the problem would have no solution. In the first target region this is ensured by the constraint of minimum acetate excretion. In the second target region we ensure this by assuming a strictly positive substrate uptake reaction ($r_s$). Constraint-based models often hold constraints that already exclude the zero-flux vector as feasible solution (e.g., a minimum non-growth associated ATP maintenance demand). If this is not the case, we must explicitly exclude the trivial flux vector, e.g. by demanding a minimum substrate uptake rate for the target region. The

second target region can thus be expressed by the three inequalities

$$r_{O2_{up}} \leq 0$$

$$r_P - Y_{min}^{P/S} \cdot r_s \leq 0$$

$$-r_s \leq -0.1$$

which can again be expressed in matrix notation:

$$\mathbf{T}_2 \mathbf{r} \leq \mathbf{t}_2$$

For the example, we also need to specify two desired regions: The first expresses that a high biomass yield, above a specified threshold $Y_{min}^{BM/S}$, must be attainable (under some minimal substrate uptake). We do not need to specify that this holds for the aerobic phase as the highest biomass yields are only possible under aerobic conditions and the solver will automatically keep some of those vectors. We thus have

$$-r_{BM} + Y_{min}^{BM/S} \cdot r_s \leq 0$$

$$-r_s \leq -0.1$$

making up the first desired system

$$\mathbf{D}_1 \mathbf{r} \leq \mathbf{d}_1$$

We then also need to ensure that, under anaerobic conditions, where high product yield is guaranteed due to the elimination of low-yield solutions in the second target region, some minimum growth is still possible. This gives rise to the second desired region for which we can use Eqs (9) and (10) in combination with the constraint for anaerobic conditions

$$r_{O2_{up}} \leq 0$$

and include them all in the compact description

$$\mathbf{D}_2 \mathbf{r} \leq \mathbf{d}_2$$

It is important to notice that the desired systems 1 and 2 cannot be jointly represented by a single desired system because their union is not convex (the same holds for the two target regions).

As a generalization, we allow in the following the definition of arbitrary many target regions, each defined by inequalities posed by an appropriate pair of matrix and vector ($\mathbf{T}_1$, $\mathbf{t}_1$), ($\mathbf{T}_2$, $\mathbf{t}_2$), . . ., ($\mathbf{T}_1$, $\mathbf{t}_1$). Likewise, we allow the definition of arbitrary many desired regions represented by ($\mathbf{D}_1$, $\mathbf{d}_1$), ($\mathbf{D}_2$, $\mathbf{d}_2$), . . .,($\mathbf{D}_k$, $\mathbf{d}_k$). In the MILP formulation (11), each target region and each desired region needs to be integrated as separate blocks in the formulation, analogous to the single regions in the original formulation. For example, an integrated MILP with two

target $(\mathbf{T_1}, \mathbf{T_2})$ and two desired $(\mathbf{D_1}, \mathbf{D_2})$ regions reads:

$$
\begin{pmatrix}
\mathbf{N}^T & \mathbf{I} & \mathbf{T_1} & & & & & \\
& \mathbf{t_1^T} & & & & & & \\
& & & \mathbf{N}^T & \mathbf{I} & \mathbf{T_2} & & \\
& & & & \mathbf{t_2^T} & & & \\
& & & & & & \mathbf{N} & \\
& & & & & & \mathbf{D_1} & \\
& & & & & & & \mathbf{N} \\
& & & & & & & \mathbf{D_2}
\end{pmatrix}
\begin{pmatrix}
\mathbf{u_1} \\ \mathbf{v_1} \\ \mathbf{w_1} \\ \mathbf{u_2} \\ \mathbf{v_2} \\ \mathbf{w_2} \\ \mathbf{r_1} \\ \mathbf{r_2}
\end{pmatrix}
\begin{matrix}
= \\ \leq \\ = \\ \leq \\ = \\ \leq \\ = \\ \leq
\end{matrix}
\begin{pmatrix}
\mathbf{0} \\ -c \\ \mathbf{0} \\ -c \\ \mathbf{0} \\ \mathbf{d_1} \\ \mathbf{0} \\ \mathbf{d_2}
\end{pmatrix}
$$

$$
\forall i \in Rev: \quad
\begin{aligned}
& z_{1,p,i} = 0 \rightarrow v_{1,i} \leq 0, && z_{1,p,i} = 1 \rightarrow v_{1,i} \geq c \\
& z_{1,n,i} = 0 \rightarrow v_{1,i} \geq 0, && z_{1,n,i} = 1 \rightarrow v_{1,i} \leq -c \\
& z_{2,p,i} = 0 \rightarrow v_{2,i} \leq 0, && z_{2,p,i} = 1 \rightarrow v_{2,i} \geq c
\end{aligned}
$$

$$
\forall i \in Rev: \quad z_{2,n,i} = 0 \rightarrow v_{2,i} \geq 0, \quad z_{2,n,i} = 1 \rightarrow v_{2,i} \leq -c
$$

$$
\forall i \in Irrev: \quad z_{1,n,i} = z_{2,n,i} = 0
$$

$$
0.25\, z_{1,p,i} + 0.25\, z_{1,n,i} + 0.25\, z_{2,p,i} + 0.25\, z_{2,n,i} \leq z_i
$$

$$
(1 - z_i) \cdot lb_i \leq r_{1,i} \leq (1 - z_i) \cdot ub_i
$$

$$
(1 - z_i) \cdot lb_i \leq r_{2,i} \leq (1 - z_i) \cdot ub_i
$$

$$
\mathbf{u_1}, \mathbf{u_2} \in \mathbb{R}^m, \mathbf{t_1} \in \mathbb{R}^{|\mathbf{t_1}|}, \mathbf{w_1} \in \mathbb{R}_{\geq 0}^{|\mathbf{t_1}|}, \mathbf{t_2} \in \mathbb{R}^{|\mathbf{t_2}|}, \mathbf{w_2} \in \mathbb{R}_{\geq 0}^{|\mathbf{t_2}|}, c > 0, \mathbf{v_1}, \mathbf{v_2}, \mathbf{r_1}, \mathbf{r_2} \in \mathbb{R}^n,
$$

$$
\mathbf{d_1} \in \mathbb{R}^{|\mathbf{d_1}|}, \mathbf{d_2} \in \mathbb{R}^{|\mathbf{d_2}|},
$$

$$
z_{1,p,i}, z_{1,n,i}, z_{2,p,i}, z_{2,n,i}, z_i \in \{0,1\}
$$

(15)

The same objective function (12) as used for single target/desired regions can be applied.

Another application of multiple target and desired regions will be presented later when searching for MCS for substrate co-feeding strategies.

## Extension 2: Weighting of interventions with cost factors

A further generalization of the MCS MILP problem is to replace the original objective function (12), which simply counts the number of interventions, with a *weighted* cost function:

$$
minimize \sum_i p_i z_i \tag{16}
$$

The weights $p_i$ can be specified by the user and reflect the experimental costs of implementing the respective intervention. For example, blocking oxygen uptake to enforce anaerobic conditions is a frequent intervention in many MCS when searching for strain designs with high product yields. Clearly, this "intervention" is experimentally easy to implement and in many biotechnological applications even the preferred process condition. The weight factor for the oxygen uptake reaction could thus be set to zero.

There is one important change in the interpretation of MCS when using the generalized objective function (16). Minimization of the original (simpler) objective function (12) (which is equivalent to set all $p_i$ to unity in (16)) always delivers MCS with a minimum number of knock-outs, hence, of MCS with minimal support. With the generalized objective function (16), the MCS still ensure that the cost function is minimized (which still justifies the use of the term "minimal" cut set). However, if some $p_i$ are set to 0, then MCS may be found where the

set of reaction knockouts is not support-minimal since some deletions may be added "for free". In those cases, there can be MCS with identical costs that differ only in the presence/ absence of these cost-free reaction deletions. It is easy to filter out equivalent MCS that are supersets of others in a postprocessing step, however, in some cases it might just be the goal to find all these alternatives. Generally, even if the $p_i$ of a reaction is zero, it might not be removable as the desired region may require the presence of this reaction.

## Extension 3: Combination of reaction deletions and additions and its application to find substrate co-feeding strategies

Most computational strain optimization algorithms use combinations of (gene or reaction) knockouts to enforce desired phenotypes. However, some dedicated methods have been developed for biased strain design techniques to also allow addition of heterologous reactions alongside reaction deletions, to further enlarge the search space for network design [8,33,34]. So far, this feature is not available in the MCS approach, but will be developed in the following. As one particular application, we will describe how substrate co-feeding strategies can be computed with this extended MCS approach.

For combining reaction deletions and additions in the MCS framework, the (new) addable reactions are included in the original network and every reaction is then marked as either "deletable", "addable" or "non-targetable". In contrast to deletable reactions, where the *knockout* of a reaction $i$ is marked by ($z_i = 1$) and implies costs in the objective function (16), addable reactions are treated inversely so that the last two equations in (11) (and analogously in (15)) for addable reactions change to:

$$z_{p,i} + z_{n,i} \leq (1 - z_i)$$
$$z_i \cdot lb_i \leq r_i \leq z_i \cdot ub_i$$

(17)

The non-use of an addable reaction $i$ is the standard case, indicated by ($z_i = 0$), while the *incorporation* of a reaction into the network is marked by ($z_i = 1$) and therefore leads to increased costs in the objective function (16).

Fig 1 illustrates that, compared to the traditional MCS approach, the generalized design specifications (multiple target and desired regions) and the extended search space (by reaction additions) enormously increase flexibility in formulating and solving complex network and strain design problems. The solid black lines mark the solution spaces of steady-state flux vectors of the wild type and of the different mutants after applying an MCS, respectively. While each target and each desired polyhedron is always convex (as are the solution spaces of the wild type and of the MCS mutant), the union of multiple *target* or *of multiple desired regions* is, in general, not.

With the extensions of multiple target regions (extension 1) and possible addition of reactions (extension 3), we are now able to compute suitable MCS that exploit substrate co-feeding, which adds another dimension for strain design strategies. Co-feeding of substrates has been used in metabolic engineering, for example, to enhance the productivity of a designed strain [35–37], to cope with genetic knockouts that induce auxotrophies [16], or to specifically design auxotrophic [38] and biosensor strains [39]. Substrate co-feeding can also be used to effectively regenerate or balance cofactors [40,41] or to provide product precursors [42]. Despite these applications, as pointed out by [42], directed strain design harnessing substrate co-feeding opportunities has rarely been employed so far. Fig 2 shows an example network holding potential for different strain design strategies that rely either on single substrate feeding or co-feeding. The main (standard) substrate S can be metabolized to biomass (BM) and four different products (P (desired product), Q, R, U). Dashed reaction arrows with yellow captions indicate

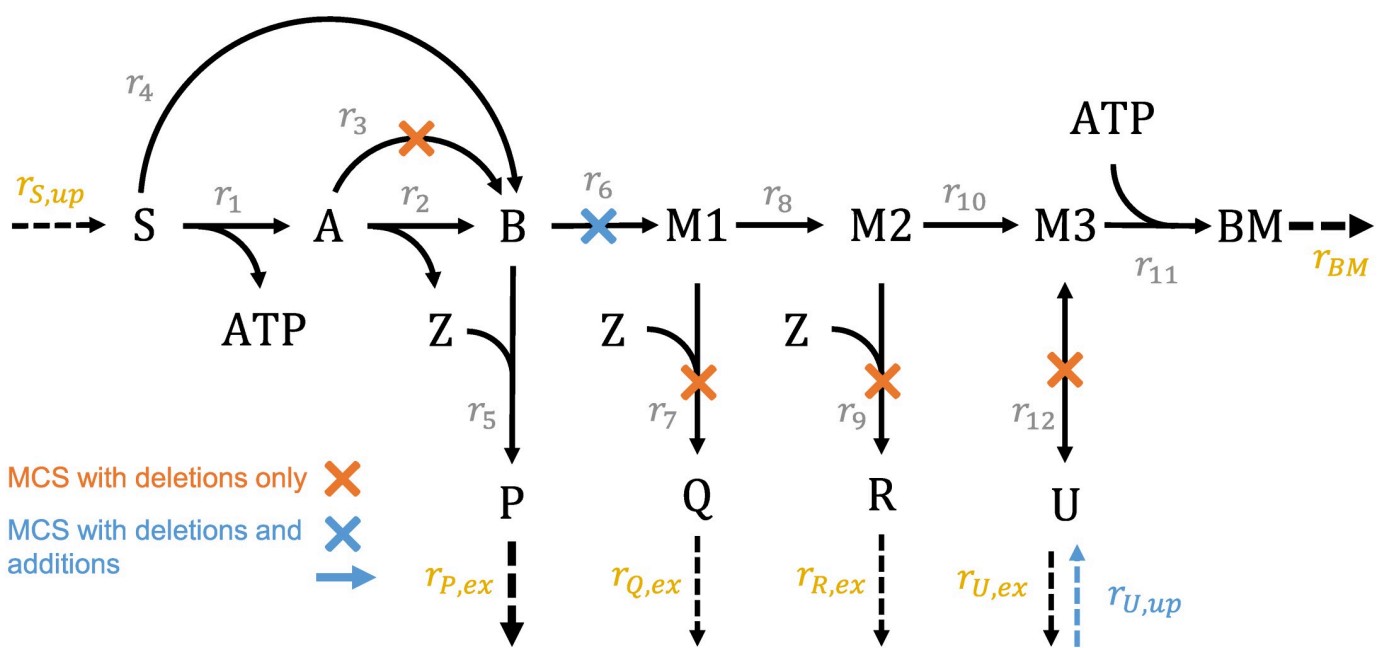

**Fig 2. Example network illustrating substrate co-utilization.** The wild-type strain that can be turned into a production host for the target product P, either by using substrate S and four knockouts (orange MCS) or by a co-utilization of S and U, which requires one deletion and the addition of the uptake reaction for U (blue MCS). Reactions with dashed arrows and yellow names indicate non-targetable exchange reactions.

exchange reactions (the latter are assumed to happen spontaneous, i.e., they are non-targetable). The blue dashed reaction indicates that U could potentially also be taken up and thus serve as a substrate. As we will see later, especially treating the case of a metabolite either being an excreted product or serving as a potential substrate is technically challenging. A realistic example would be acetate that can often be excreted as product but may also serve as relatively cheap (co-)substrate in bioprocesses. We assume that P is the product of interest. As a typical strain design strategy we may enforce growth-coupled product synthesis where the target region contains flux vectors with low product yield (e.g., $Y_{P/S} = \frac{r_{P,ex}}{r_{S,up}} \leq 0.4$; see Eq (7)), while the desired region demands that biomass synthesis is feasible (e.g., $r_{BM} \geq 0.1$; see Eq (4)). A traditional MCS-based strategy delivered by the algorithm is indicated in Fig 2 (knockouts marked in orange): S is used as the sole substrate and $r_3$, $r_7$, $r_9$ and $r_{12}$ are deleted. In the remaining network, ATP production entails the generation of Z, a metabolite that can only be drained by synthesizing and excreting P. The option to co-feed metabolite U offers now another coupling strategy by dissecting the metabolism into an upper and a lower part (interventions marked in blue in Fig 2). The removal of $r_6$ and the addition of the uptake reaction for U ($r_{U,up}$) leads to a functional separation of the two substrates: S is then used for ATP and product synthesis while U is needed as biomass precursor.

Generally, scenarios with multiple substrates require a suitable redefinition of the yield to properly account for both substrates. In our example we would consider the combined yield $Y_{P/(S+U)} = \frac{r_{P,ex}}{r_{S,up}+r_{U,up}}$ and specify the target region by $Y_{P/(S+U)} \leq 0.4$. We could then mark $r_{U,up}$ as addable reaction and try to search for both classical one-substrate strain designs (using only S) as well as substrate co-feeding strategies (using S and U). If U would only act as optional substrate and the excretion of U was not possible, then co-feeding strategies could be straightforwardly identified by marking the uptake of U as addable reaction. However, in this example

with U also being a potential product, this alone would not be sufficient to find suitable co-feeding solutions. The MCS algorithm would seek to eliminate all flux states with yields inferior to $Y_{P/(S+U)}$. Yet, the cycle of the sink pseudo-reaction $r_{U,ex}$ and an activate uptake reaction $r_{U,up}$ can carry high fluxes and inflate the denominator of the yield function, making it impossible to disrupt all flux distributions with yields below the threshold. Allowing the deletion of $r_{U,ex}$ would affect the correct identification of single substrate strategies because the algorithm would then be able to generate solutions that delete the secretion reaction $r_{U,ex}$ although it is non-targetable.

A solution to this dilemma is the definition of two target regions. The first target polyhedron describes undesired solutions for the "classical" one-substrate usage where U is not taken up:

$$\mathbf{T1}: \quad Y_{P/S} = \frac{r_{P,ex}}{r_{S,up}} \le 0.4, \quad r_{S,up} \ge 0.1 \frac{mmol}{gDW\ h}, \quad r_{U,up} = 0 \frac{mmol}{gDW\ h} \tag{18}$$

These constraints can be reformulated to bring them into the required form $\mathbf{T_1 r} \le \mathbf{t_1}$. The second target region specifies undesired solutions for the case where U is taken up, using the combined yield term. Importantly, in the definition of this second target region, only the relevant cases where U is not excreted are considered. Hence, with the defined target regions 1 and 2, the irrelevant case with an active cycle in $r_{U,up}$ and $r_{U,ex}$ is simply not included in the set of target vectors:

$$\mathbf{T2}: \quad Y_{P/S,U} = \frac{r_{P,ex}}{r_{S,up} + r_{U,up}} \le 0.4, \quad r_{S,up} \ge 0.1 \frac{mmol}{gDW\ h}, \quad r_{U,ex} = 0 \frac{mmol}{gDW\ h} \tag{19}$$

Together with the desired region ($r_{BM} \ge 0.1$), the enumeration of strain designs with the introduced MCS algorithm extended with multiple target regions and addition of reactions will return both solutions (blue and orange in Fig 2). This MCS computation example is also available at GitHub (see below).

## Extension 4: Compression of GPR rules for computing MCS with genetic interventions

The MCS approach was originally constructed to find suitable combinations of *reaction* deletions that block a certain metabolic phenotype. However, when using it for strain design in metabolic engineering, in most cases only *genes* can be targeted directly (exceptions are pseudo-reactions for uptake of oxygen or substrates, which can be switched off by removing the respective compound from the medium). A translation of reaction deletions to corresponding gene deletions is often possible, however, there are also complicated relationships, where this translation is not straightforward, for example, when an enzyme or enzyme subunit is involved in several reactions. Hence, for the application as strain design tool, MCS should deliver suitable combinations of gene interventions (gene MCS), which induce the desired metabolic phenotype by indirectly targeting the operation of certain reactions [25]. This requires the incorporation of gene-protein-reaction (GPR) rules, which has been done in several strain optimization methods [7,18,25,26,43,44]. GPR rules are typically Boolean functions in disjunctive normal form (DNF) describing how the enzymes catalyzing the metabolic reactions in the network are made from combinations of gene products. Two major approaches for integrating GPR rules in the MCS framework have been suggested [25,26]. Herein we adopt the approach of Machado et al. [25] where the GPR rules are translated to pseudo-reactions and pseudo-metabolites, which are seamlessly integrated into the metabolic network. The reaction-based MCS algorithm can then remove or add genes via their associated

**Fig 3. Integration of GPR rules into the metabolic network structure.** In contrast to the approach of Machado et al. [25], in the variant introduced herein, every reaction with a GPR rule consumes a pseudo-metabolite Q that represents a pool of enzymes catalyzing this reaction. Example cases: A) reversible reactions; B) enzymes with subunits; C) promiscuous enzymes ($g_1$) and isoenzymes ($g_1, g_2$).

pseudo-reactions. In the original approach of Machado et al., each gene product (i.e., either an enzyme or an enzyme subunit denoted by $E$, $E_1$, $E_2$ ... in Fig 3) is introduced as a pseudo-metabolite that is generated by an individual "*enzyme synthesis*" pseudo-reaction (denoted by $g_1$, $g_2$ ... in Fig 3) from its corresponding gene. (Note that, depending on the context, we will use the symbols $g_1$, $g_2$ ... interchangeably either as the enzyme synthesis pseudo-reaction or as the actual gene from which the enzymes are made. This is justified because there is a 1:1 relationship between both.) Each enzyme pseudo-metabolite $E$, $E_1$, $E_2$ ... is then integrated as a reactant in the respective reaction(s) it catalyzes and it thus becomes essential for the operation of the(se) reaction(s). Reversible reactions need to be split and considered as separate reactions for both directions. Depending on the GPR rules, it can happen that (1) one enzyme catalyzes multiple reactions, (2) a reaction requires several enzyme subunits, or (3) one reaction is catalyzed by multiple (iso)enzymes (see Fig 3). In the latter case, Machado et al. split the reaction to account separately for each isoenzyme [25]. A disadvantage of this approach is that copying an isozyme reaction with a finite and non-zero flux bound may result in an effective reaction flux that is multiple times higher than intended by the original model (see Fig 3C). We therefore choose a slightly different representation of the GPR rules, which guarantees that the flux ranges from the original model are conserved for all reactions (see right column in Fig 3). The

key difference of this approach is that we introduce for each reaction a separate pseudo-metabolite (Q) representing the pool of all enzymes (including those arising by the combination of enzyme subunits) that can catalyze this reaction. This reaction-specific enzyme pool can be filled by pseudo reactions ($p_1, p_2 \ldots$ in Fig 3) according to the respective GPR rules. The enzyme pool Q is then "consumed" in its associated reaction. Hence, reactions are only split in the case of reversibility, where the flux boundaries can easily be mapped from the original reaction. Auxiliary pseudo-reactions are all unidirectional and unbounded. Our approach adds more pseudo-metabolites and pseudo-reactions than the original approach, however, many of these additional elements disappear again when compressing the network with integrated GPR rules (see below).

Once the GPR rules have been added to the metabolic network, the MCS algorithm can be used to generate gene intervention based strain designs. Here, metabolic reactions remain protected from knockouts (non-targetable) while the manipulation of the pseudo-reactions yielding the enzymes or enzyme subunits from the genes (reactions $g_1, g_2 \ldots$ in Fig 3) is allowed. Reactions such as oxygen or substrate uptake, which have no associated genes but can be controlled externally, can also be kept targetable. In this way, the MCS algorithm can identify suitable combinations of genetic interventions and process conditions to meet the strain design specifications.

The extension of metabolic networks with GPR rules as described above may significantly increase the network size and thus the complexity of strain design methods. Generally, compression techniques and preprocessing steps to cope with the complexity of (classical) genome-scale MCS calculations are essential and integrated in some strain design (including MCS) algorithms. These methods include, for example, the removal of conservation relations and blocked reactions as well as lumping of coupled reactions (see e.g. [30,45]). In addition to these network compression techniques already in use, we here propose a set of preprocessing steps to also reduce and simplify GPR rules before integrating them in the metabolic network. For gene-based MCS, every gene is simulated as a targetable reaction that has an associated Boolean variable in the underlying MILP indicating whether it is active or not. Boolean variables are computationally more expensive than continuous variables. The compression steps introduced below seek to reduce the number of targetable reactions (here corresponding to targetable genes) without losing any MCS solution.

A first step for the reduction of network and GPR rules is the identification and removal of blocked reactions and their genes. More advanced compression steps described below exploit the fact that the *desired region(s)* render several reactions essential. This can be used to discard certain targetable genes from the model as their knockout would block essential or at least never affect non-essential reactions. For example, if some minimum growth rate has to be maintained in the designed strains, any gene being essential for growth needs not be considered as knockout candidate and can be removed from the GPR rules. In practice, essential reactions can be identified by flux variability analysis (FVA), performed on each desired region. If, for any desired region, upper and lower bound of a reaction are non-zero and have the same sign, then the reaction is essential and must not be blocked by any set of gene deletions. Apart from essential genes, our compression steps introduced below also exploit the occurrence of "equivalent genes" which can be lumped. In total, we use the following seven compression steps to reduce the number of genes and GPR rules added to the model:

1. Remove GPR rules of blocked reactions. Deletion (or addition) of genes will have no effect on these reactions, hence their rules are not relevant for the MCS computation.

2. Mark a gene $g_p$ as protected ($g_p = 1$) and thus non-targetable if it occurs exclusively in GPR rules for essential reactions. Knock-out of such a gene can never contribute in blocking non-essential reactions.

3. Mark a gene as protected if it is essential for at least one essential reaction since deleting this gene will inevitably result in disrupting the desired behavior (note: rule 3 differs from rule 2 as the gene in rule 2 might not be essential).

4. Identify GPR terms (disjunctions) that cannot be targeted due to protected genes. If one of the conjunctions of a DNF consist exclusively of protected genes, then the respective reaction cannot be knocked out by any combination of gene deletions (e.g., if gene $g_p$ is protected ($g_p = 1$) then $g \bigvee g_p = 1$). This makes the GPR rule irrelevant for MCS computation and it can be discarded.

5. Discard all protected genes. After step 4, all protected genes occur in conjunctions with unprotected genes and their consideration is no longer necessary ($g_p = 1 \rightarrow g \bigwedge g_P = g$).

6. Reduce non-minimal conjunctions in the GPR rule that arose by the previous steps (e.g., $g_1 \bigvee (g_1 \bigwedge g_2) = g_1$).

7. Lump equivalent gene deletion candidates that always occur together in conjunctions (e.g., $g_1 \bigwedge g_2$) and lump gene addition candidates likewise. In some cases, another lumping can be performed inside single GPR rules if the genes do not occur in any other context. Finally, repeat the substitution also for the disjunctions of type ($g_{1 \wedge 2} \bigvee g_3$). To map the original intervention costs, each lumped gene carries the minimum intervention costs of the gene rule that was substituted. Computed MCS involving lumped genes must be expanded in a post-processing step.

We added the GPR rule compression and integration as a building block in the pipeline for MCS computation. This pipeline now involves problem setup, GPR rule compression and integration, network compression, and post-processing steps for expanding computed MCS (see also the example in Figs 4 and 5 discussed below):

(0). Definition of network, target and desired regions(s), and of addable/deletable/non-targetable reactions/genes.

(1). FVA of the full model to find blocked reactions and FVA of the desired regions to identify essential reactions

(2). Compression of GPR rules (making use of the FVA results from step (1))

(3). Integration of compressed GPR rules in the network

(4). Network compression (removal of blocked reactions and conservation relations, lumping reactions, protection of essential reactions, determination of effective flux bounds etc.)

(5). MCS computation using the MCS core algorithm

(6). Decompression of lumped reactions

(7). Decompression of lumped genes

Note that the steps (4)-(6) (leaving out steps (1)-(3) and (7)) can still be used to compute classical MCS with reaction knockouts. In *CellNetAnalyzer* (see below), these steps are conducted by a function called CNAMCSEnumerator2. The steps (1)-(3) and (7) are specific for the calculation of gene MCS and in *CellNetAnalyzer* they are performed by the function

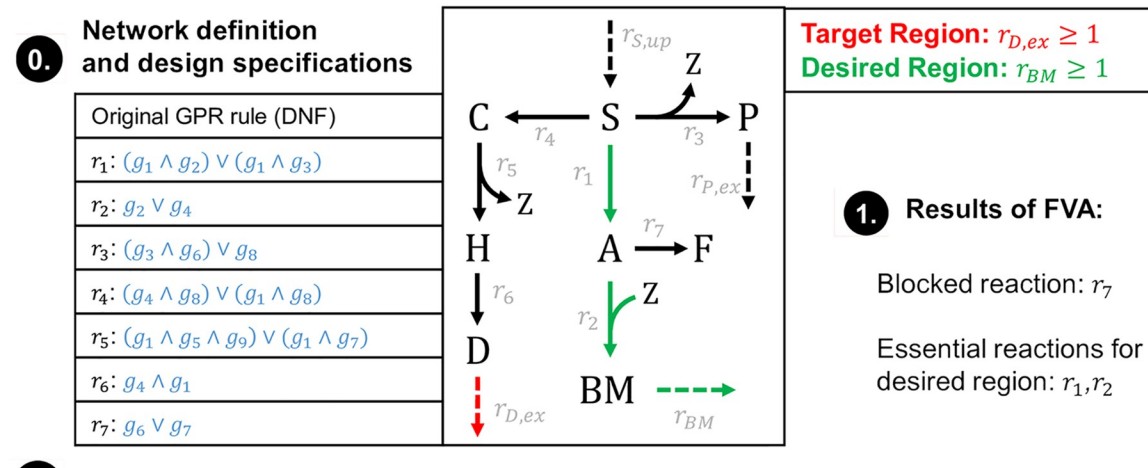

## 2. Compression of GPR rules

|  | 1) Remove blocked reactions ($r_7$) 2) Protect genes that occur only in essential reactions ($g_2$) | 3) Protect genes that are essential for essential reactions ($g_1$) | 4) Discard GPR rules that cannot be knocked out 5) Discard protected genes | 6) Remove non-minimal GPR rules 7) Lump equivalent genes |
|---|---|---|---|---|
| $r_1$ | $(g_1 \wedge \boldsymbol{g_2}) \vee (g_1 \wedge g_3)$ | $(\boldsymbol{g_1} \wedge \boldsymbol{g_2}) \vee (\boldsymbol{g_1} \wedge g_3)$ |  |  |
| $r_2$ | $\boldsymbol{g_2} \vee g_4$ | $\boldsymbol{g_2} \vee g_4$ |  |  |
| $r_3$ | $(g_3 \wedge g_6) \vee g_8$ | $(g_3 \wedge g_6) \vee g_8$ | $(g_3 \wedge g_6) \vee g_8$ | $g_{3 \wedge 6} \vee g_8$ |
| $r_4$ | $(g_4 \wedge g_8) \vee (g_1 \wedge g_8)$ | $(g_4 \wedge g_8) \vee (\boldsymbol{g_1} \wedge g_8)$ | $(g_4 \wedge g_8) \vee g_8$ | $g_8$ |
| $r_5$ | $(g_1 \wedge g_5 \wedge g_9) \vee (g_1 \wedge g_7)$ | $(\boldsymbol{g_1} \wedge g_5 \wedge g_9) \vee (\boldsymbol{g_1} \wedge g_7)$ | $(g_5 \wedge g_9) \vee g_7$ | $g_{(5 \wedge 9) \vee 7}$ |
| $r_6$ | $g_4 \wedge g_1$ | $g_4 \wedge \boldsymbol{g_1}$ | $g_4$ | $g_4$ |
| $r_7$ | $\cancel{g_6 \vee g_7}$ |  |  |  |

## 3. Network extension with compressed GPR rules

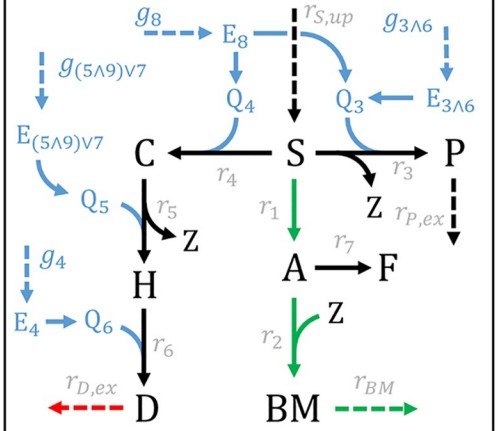

## 4. Network compression

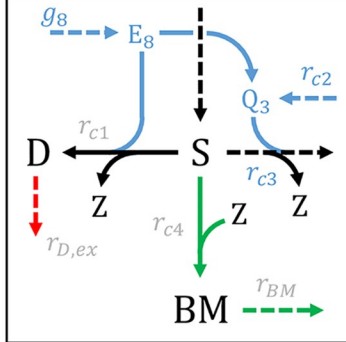

$r_{c1}: g_{(5 \wedge 9) \vee 7}, g_4, r_4, r_5, r_6, (p_{g4 \rightarrow Q6}, p_{(g5 \wedge g9) \vee g7 \rightarrow Q5})$
$r_{c2}: g_{3 \wedge 6}, (p_{g3 \wedge g6 \rightarrow Q3})$
$r_{c3}: r_3, r_{P,ex}$
$r_{c4}: r_1, r_2$

**Fig 4. Example illustrating the 7 steps of gene MCS calculation.** Red arrow: targeted reaction; green arrows: protected reactions; blue: GPR rules; dashed blue arrows: switchable gene pseudo-reactions. For the sake of readability, the protein pool pseudo-reactions ($p$) connecting the enzymes ($E$) with the enzyme pools ($Q$) were not labeled in the network in step 3 and 4. The reactions $r_{c1} \ldots r_{c4}$ represent lumped reactions resulting from network compression (the contained reactions are shown below the box of step 4). The subsequent steps 5–7 of this example are shown in Fig 5.

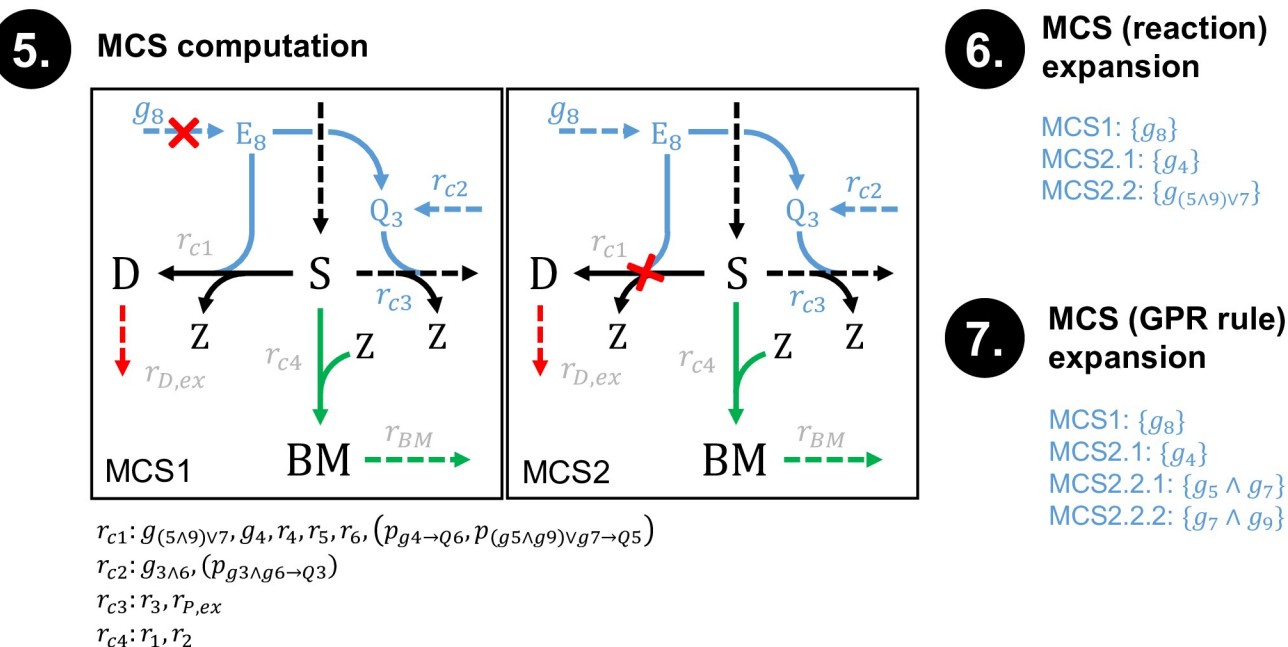

**Fig 5. Continuation of the exemplary gene MCS calculation from Fig 4.** Shown are the three final steps of MCS computation (step 5), expansion/ decompression of MCS at the reaction (step 6) and then at the gene (step 7) level. For better readability, the protein pool pseudo-reactions (*p*) connecting the enzymes (E) with the enzyme pools (Q) were not labeled in the network in step 5.

CNAgeneMCSEnumerator which calls the CNAMCSEnumerator2 between step (3) and (7) as a sub-module. In this way, the actual gene MCS are computed as reaction knockouts of the enzyme synthesis reactions as explained above.

Figs 4 and 5 show an example of the whole pipeline demonstrating how the compression of the GPR rules and subsequently of the network structure reduces the dimension of the problem. In the presented example network, the substrate S can be turned into biomass and the two products P and D. The strain design goal is to inhibit the production of D by suitable genetic knockouts, while maintaining the ability to grow. An FVA shows that the reactions $r_1$ and $r_2$ are essential for the desired flux states. This information is used to compress the original GPR ruleset along the presented steps (Fig 4). The reconfigured network with integrated compressed GPR rules is presented in Fig 4, step 3, showing that from the nine genes only four (partially lumped) genes remain in the network after GPR compression. Next, classical network compression (Fig 4, step 4) lumps coupled reactions, not distinguishing between "real" metabolic reactions and GPR pseudo-reactions. Analogous to lumped gene interventions, for each lumped reaction (like $r_{c1}$, $r_{c2}$ and $r_{c3}$ obtained in step 4), an individual cost-factor is generated that equals the minimal intervention cost for the ensemble of the original reactions.

Applied to the compressed network, the MCS algorithm will deliver two solutions to suppress the undesired behavior: either the knockout of $g_8$ or of the lumped reaction $r_{c1}$ (Fig 5, step 5). In order to match the original model, these solutions need to be expanded by reversing the previous compression steps, first at the reaction (Fig 5, step 6) and then at the gene level (Fig 5, step 7). In the example, decompressing the MCS results in four final solutions: A knockout of $g_8$, a knockout of $g_4$ or a knockout combination of either $g_5$ and $g_7$ or $g_7$ and $g_9$.

As shown in this example, decompression of lumped genes and reactions will not only result in a higher number of MCS, it may also lead to a higher number of interventions per MCS or to more expensive MCS. Yet, the cost-reattribution during GPR compression step 7

ensures that any MCS found in a compressed network still expands to at least one MCS in the original network with identical costs (e.g. $g_{((5 \wedge 9) \vee 7)}$ would have deletion costs of 2 in the compressed network). In more complex cases where MCS have multiple expansions with different costs, all MCS that exceed the MCS cost limit are discarded.

### Implementation and availability

All scripts and network files needed to reproduce the example calculations from Figs 2 and 4, and 5 as well as the calculations performed in the Results section are provided at GitHub: https://github.com/ARB-Lab/MCS_extended. The scripts can be run with the newest version (2020.2) of the freely available MATLAB toolbox *CellNetAnalyzer* [46,47], where the new features described herein have been integrated in API functions. For all benchmarking and genome-scale MCS computations presented in the Results section we used MATLAB 2019b and IBM ILOG CPLEX 12.10 on a high-performance cluster. Each computation was performed on single nodes with two 8-core Intel Xeon Skylake Silver 4110 and 192 GB memory.

## Results

In the Methods section we introduced several generalizations and new algorithmic developments for the computation of MCS, which are especially (but not exclusively) useful for strain design applications. In the following, we will demonstrate the applicability and the performance benefits of these extensions by realistic example calculations of relevant *E. coli* strain designs for production of 2,3-butanediol (2,3-BDO). 2,3-BDO is a bulk chemical whose biobased production received much attention in the field of metabolic engineering [48,49]. The spectrum of its industrial application covers a broad range from cosmetics and nutrition to bioplastic [50]. Previous studies could increase the productivity of natural producers like *Klebsiella pneumonia* or *Bacillus licheniformis*, however, the use of these species has several disadvantages due to requirement of complex and expensive medium components and potential pathogenicity [51]. Other studies therefore aimed to synthesize 2,3-BDO with more established production organisms such as *E. coli* via heterologous pathways [51–53]. For the computation examples presented below, we therefore consider *E. coli*, equipped with the heterologous linear pathway for 2,3-BDO synthesis via α-acetolactate and R-acetoin [51] and computed different sets of MCS to design strains with growth-coupled 2,3-BDO synthesis. Using different scenarios, we first benchmarked the performance gain in the MCS computation through our introduced compression rules for GPR associations against the conventional approach in both a core and a genome-scale model of *E. coli*. Afterwards, we exemplify the use of multiple target and desired regions and the possibility to consider combinations of gene deletions and additions to obtain different strain designs including substrate co-feeding strategies for 2,3-BDO synthesis.

### Effect of GPR rule compression on MCS computation performance

MILPs used in the context of different strain design techniques are often very large and complex, especially when applied in genome-scale networks. Loss-free compression techniques may enhance the performance of these MILPs and are often even essential to reach acceptable runtimes and to obtain a significant pool of solutions. Network compression has become a standard tool in many constraint-based calculations [30,45], including MCS approaches [18]. In addition, we herein introduced a set of rules for compressing GPR associations, which are relevant when metabolic interventions are directly calculated at the gene instead of the reaction level. Thus, GPR and network compression tackle different model redundancies.

**Table 1. Benchmark results of different compression scenarios in two MCS computation setups (core and genome-scale model of *E. coli*) for growth-coupled 2,3-BDO synthesis.** The rows "#Genes" and "#GPR rules" refer to the respective number of genes and GPR rules eventually integrated in the network after GPR rule compression (as long as the latter is conducted; otherwise it refers to the original number of genes and GPR rules). Likewise, the rows "#Species", "#Reactions","#Targetable reactions/genes" refer to the numbers of species, reactions, and targetable genes/reactions, respectively, after integration of the (compressed/non-compressed) GPR rules and/or after network compression.

| | Target: $Y_{23BDO/Glc} \leq 0.3 * Y_{23BDO/Glc,max}$, Desired: $\mu \geq 0.05$ h$^{-1}$ | | | | | | | |
|---|---|---|---|---|---|---|---|---|
| | EColiCore2 (max MCS size: 9) | | | | iML1515 (max MCS size: 7) | | | |
| | 502 reactions, 486 species, 508 genes, 683 GPR rules | | | | 2715 reactions, 1879 species, 1516 genes, 3828 GPR rules | | | |
| | No compression | Network compression | GPR compression | Network + GPR compression | No compression | Network compression | GPR compression | Network + GPR compression |
| #Genes (after compression) | 508 | 508 | 114 | 114 | 1516 | 1516 | 649 | 649 |
| #GPR rules | 683 | 683 | 163 | 163 | 3828 | 3828 | 1531 | 1531 |
| #Species | 1435 | 244 | 682 | 107 | 5616 | 1207 | 3529 | 1068 |
| #Reactions | 1690 | 500 | 799 | 233 | 8103 | 2513 | 4995 | 2245 |
| #Targetable reactions/genes | 251 | 181 | 64 | 58 | 1320 | 631 | 574 | 528 |
| #MCS found | 6025 | 92 | 169 | 89 | **Not finished (>100h)** | 177 | 190 | 177 |
| #MCS found after decompression | 6025 | 6025 | 6025 | 6025 | | 2632 | 2632 | 2632 |
| Runtime [min] | **5199.7** | **178.9** | **41.0** | **11.5** | | **707.8** | **846.9** | **180.3** |
| Pre- and post-processing overhead [min] | 3.8 | 6.4 | 8.4 | 6.2 | | 18.8 | 12.3 | 11.7 |

To benchmark the effectiveness of GPR and/or network compression, we compare the impact of both compression routines on the final problem size, the MILP runtime and the computational overhead in a small-scale (a slightly adapted version of *EColiCore2*, a model of the central metabolism of *E. coli* presented by Hädicke and Klamt (2017) [54]) as well as in a genome-scale (*iML1515* model by Monk et al. (2017) [55]) MCS computation setup (Table 1). Within these two models, which were both extended with GPR rules as described in the Methods section, we fully enumerate gene MCS up to the size of 9 (core) and 7 (genome-scale) gene knockouts to obtain growth-coupled strain design that produce 2,3-BDO from glucose with a minimum product yield of at least 30% of the theoretical (stoichiometric) maximum under a minimum attainable growth rate of 0.05 h$^{-1}$. As explained in the Methods section, these constraints can be expressed by one target and one desired region. We assume that all genes are targetable. Furthermore, the oxygen uptake reaction remains targetable so that the algorithm may also find anaerobic strain designs. Although (non-exhaustive) sampling of single or multiple MCS is possible and sometimes preferred in practice [18], we here perform a full enumeration up to the given maximum MCS size to ensure that identical pools of solutions are delivered for the different runs. With the specified target and desired regions, there exist 6025 solutions (MCS) in the core and 2632 solutions in the genome-scale setup (Table 1).

A comparison of the MILP runtimes for the four different runs (with/without GPR rule compression and with/without network compression) reveals the effectiveness of the state-of-the-art network compression procedure but also demonstrates the advantages of an extended compression by the herein presented GPR compression procedure. In fully uncompressed models, the computation time for MCS lies in the range of days even in the core model while genome-scale computation is not possible at all. The benchmarked scenarios show that at least one sort of problem compression, network or GPR rule, is necessary to achieve reasonable runtimes of the MCS algorithm. The different compression steps reduce the number of reactions and metabolites by up to more than 90%, which in turn also leads to an enormous

reduction in the number of MCS found in the compressed network (e.g., 89 vs. 6025 in the core network), which are expanded to the full set of MCS in a post-processing step. Each MCS of the compressed network is a representative of an MCS equivalence class (see [56]). As expected, MCS computation with combined network and GPR compression has superior performance: compared to the traditional pure network compression it runs about 15 times faster in the core setup and reduces the runtime of the genome-scale setup by a factor of about 4. The overhead of the pre- and post-processing steps, which includes network and GPR compression, is comparably small, especially in the genome-scale network (<7%), and thus more than justified. The additional speedup achieved by the GPR rule compression is vital for genome-scale computation of gene MCS, since the runtimes can range from hours to days and up to weeks. It should also be noted that we considered all 1516 genes in the genome-scale model as potential targets, in contrast to many other studies where the set of targetable genes (or reactions) is often manually reduced to shrink the problem size to a manageable dimension.

Next we compared our approach for calculating MCS with integrated GPR associations with the gene cut set (gMCS) approach of Apaolaza et al. [26,27]. The latter determines in a preprocessing step a mapping of minimal gene knockout sets required to block at least one reaction. As the gMCS method can so far only deal with target reactions but not with desired regions we used the calculation of synthetic lethals (SL; combinations of reaction knockouts that block growth) in the genome-scale iML1515 network as benchmark. We enumerated all SL up to size 4 and found with both methods the identical set of 889 MCS which confirms their consistency. For the runtimes we determined 65 minutes with our method and 163 minutes with gMCS. The scripts of these calculations are also available at the provided GitHub link.

## Using new MCS features for strain design

Next, we used the 2632 strain designs found for 2,3-BDO synthesis from glucose in the genome-scale model as a reference to compare it with the new features of the MCS framework such as multiple target and desired regions, gene/reaction additions and substrate co-feeding. The different computation setups are listed in Table 2. As a first extended setup, we constrained the genome-scale search of MCS for growth-coupled 2,3-BDO synthesis by introducing a second desired region that requires any mutant to cope with an increased ATP maintenance rate ($r_{ATPM} \geq 18\ mmol\ g_{BDW}^{-1}\ h^{-1}$; scenario 2 in Table 2). This supplementary constraint could be used for developing more stress resistant strains or to prime strains for ATP-wasting boosted production [57–59]. The enumeration of gene MCS for this problem up to size 7 returned 544 strain designs, expanded from 25 MCS found by the MILP in the compressed network. As expected, for reasons of consistency, all 544 MCS were contained in the

**Table 2. Different genome-scale MCS computation setups for growth-coupled 2,3-BDO synthesis.**

|   | Scenario | Model | Target regions | Desired regions | Max. MCS size/cost | Runtime [h] | Solutions |
|---|----------|-------|----------------|-----------------|--------------------|-------------|-----------|
| 1 | Standard (cf. Table 1) | iML1515 | 1 | 1 | 7 | 3.0 | 2632 (S1 Table) |
| 2 | High ATP maintenance | iML1515 | 1 | 2 | 7 | 3.3 | 544 (S2 Table) |
| 3 | Multiple substrates | iML1515 | 2 | 1 | 6 | 35.3 | 2611 (S3 Table) |
| 4 | Multiple substrates + high ATP maintenance | iML1515 | 2 | 2 | 6 | 17.3 | 995 (S4 Table) |

previously found set of 2632 MCS. To ensure that the found solutions were not only correct but also complete, we filtered the 2632 original MCS (from Table 1; see also scenario 1 in Table 2) by their compliance with the ATP maintenance constraint and obtained an identical set of 544 MCS. Interestingly, all these MCS with increased ATP production capabilities use aerobic conditions, while all remaining MCS involve the removal of the oxygen supply.

To validate the operability of MCS searches also with multiple target regions and with reaction/gene additions, we included options to use alternative substrates or to use co-feeding of multiple substrates (Table 2, scenario 3). We extended the network with uptake reactions for acetate and glycerol and tagged them as addable, together with glucose. Accordingly, the algorithm now finds MCS that use glucose, glycerol, or acetate or any combination of these. We adapted the previously defined yield constraint to account for all three substrates. The individual weight of the substrates (and the product) in the yield function was determined by the number of carbon atoms in the corresponding molecules. As a result, the yield constraint expresses a minimum threshold for the carbon recovery of the combined substrates in the final product. Acetate, however, takes on a dual role in this scenario because some strain designs might require it as a co-substrate while others require its secretion as a by-product. To account for this equivocalness, the minimum yield constraint was expressed by two target regions, as described in the Methods section, one for acetate by-production (Eq (20)) and one for the case when acetate is used as a (co-)substrate (Eq (21)):

$$\textbf{T1}: \quad 4\ r_{23-BD,ex} + 0.3 \cdot \frac{4}{6} Y^{max}_{23-BDO/glc}\ \left(6\ r_{glc,up} + 3\ r_{glyc,up}\right) \le 0\ , \ r_{ac,up} = 0 \tag{20}$$

$$\textbf{T2}: \quad 4\ r_{23-BDO,ex} + 0.3 \cdot \frac{4}{6} Y^{max}_{23-BDO/glc}\ \left(6\ r_{glc,up} + 2\ r_{ac,up} + 3\ r_{glyc,up}\right) \le 0\ , \ r_{ac,ex} = 0 \tag{21}$$

Note that in the specification above, we follow the definition of iML1515 where the exchange reactions are positive if they export a compound and negative if they import it. The carbon yield threshold was again set to 30% of the maximum stoichiometric carbon yield, referring to glucose as the substrate. This definition of the yield threshold ensures that the subset of the found MCS that rely solely on glucose is directly comparable to the previously found 2632 MCS. In the latter case, the carbon-based yield formulation is equivalent to the previously used molar yield constraint. In the same manner, the desired region was now specified as biomass yield per carbon-weighted substrate:

$$\textbf{D1}: \quad - r_{BM} - 0.005 \cdot \frac{1}{6}\ \left(6\ r_{glc,up} + 2\ r_{ac,up} + 3\ r_{glyc,up}\right) \le 0 \tag{22}$$

Again, an additional specification of the requested minimal 2,3-BDO yield is not necessary because all (target) regions with lower product yields will be eliminated by the MCS while the desired Region D1 ensures that other flux vectors, then with high product yield, still exist.

The described scenario with multiple substrates leads to a largely extended solution space of feasible flux vectors, which, together with the more complex description with two target regions, render the MCS computation significantly more complex. For this reason, we enumerated MCS only up to the cost of 6. In this scenario, gene knockouts were attributed with a cost factor of unity while the addition of substrates and the removal of the oxygen supply reaction remained "free" (zero costs).

The MILP for finding MCS for growth-coupled synthesis of 2,3-BDO on multiple substrates returned 574 solutions in the compressed network. Removing the redundant (non-minimal) cut sets resulted in a solution pool of 184 MCS which were then expanded to 2611 MCS (see scenario 3 in Table 2). This pool contained 2112 MCS with glucose as the sole substrate, 24

MCS relying on glycerol alone, and 475 MCS that utilize glucose with co-feeding of acetate (these solutions are further discussed below). As a proof of consistency, it was verified that the 2112 MCS with glucose as sole substrate are all contained in the pool of 2632 MCS from the initial (benchmark) setup (the additional 520 MCS require 7 gene knockouts whereas only 6 were allowed in the multiple substrates scenario).

In the last scenario 4 (Table 2) we extended the previous setup for multiple substrates (with two target regions and one desired region) by a second desired region that demands the protection of high ATP maintenance rates as previously used in scenario 2 ($r_{ATPM} \geq 18 \; mmol \; g_{BDW}^{-1} \; h^{-1}$). This computation returned 955 non-redundant MCS. Among these solutions were 24 purely glucose-based MCS, which are identical to the 24 MCS of size 6 contained in the MCS solution pool of scenario 2 in Table 2, again confirming correctness and completeness. Another set of 24 MCS, solely based on glycerol as the substrate, is identical to the set of glycerol MCS found in scenario 3 with two target and one desired region. The other 947 of the 995 MCS were supersets of the formerly found MCS in scenario 2. They nevertheless represent truly distinct and minimal intervention strategies for this particular scenario: all of these MCS add the supply of glycerol and/or acetate to satisfy the higher ATP maintenance rate. 208 of these MCS contain further cuts because the expansion of the solution space by the additional substrates would otherwise render the target region feasible again and thus allow lower product yields. All MCSs computed in the four scenarios in Table 2 have been ranked according to the criteria proposed in [60] and are provided in the S1–S4 Tables. In the following we discuss major principles of the found strain designs with or without substrate co-feeding in scenarios 1–3.

## Key principles of the found intervention strategies for 2,3-BDO synthesis

The pool of the found MCS with or without substrate co-feeding can be divided into three major classes, according to their growth-coupling mechanisms (Fig 6). 2088 of the 2632 MCS with glucose as sole substrate use anaerobic conditions (Fig 6, red) establishing growth coupling by a suitable combination of redox balancing and exclusion of alternative carbon drains. Since the 2,3-BDO pathway (consuming one NADH) is imbalanced with glycolysis (yielding two NADH), these anaerobic strategies require at least one additional by-product to balance the cell's redox state (e.g. succinate). However, these MCS must then also ensure that only a limited amount of the substrate can run to this additional side product since otherwise only a low yield of 2,3-BDO would be achieved. The other 544 of the 2632 MCS found in scenario 1 (with glucose as substrate) favor aerobic conditions (Fig 6, blue) and use oxygen as electron acceptor. In contrast to the anaerobic strategies, *all* pathways to alternative fermentation products as well as parts of the respiratory pathway (e.g., ATP synthase) must be blocked to prevent the complete oxidation of glucose to $CO_2$ in the presence of oxygen. When acetate is available as a co-feeding substrate (scenario 3), the extended MCS algorithm was able to identify completely anaerobic strain designs where redox balance could be established by reducing the provided electron acceptor acetate to ethanol (Fig 6, shown in green). Furthermore, it was able to identify alternative strain designs that replace glucose with glycerol as the sole carbon and energy source under aerobic conditions (S3 Table).

## MCS for growth-coupled 2,3-BDO synthesis in two other genome-scale models

To finally demonstrate that our extended MCS approach also works with other metabolic networks, including an eukaryotic one, we repeated all calculations for growth-coupled 2,3-BDO synthesis with *E. coli* (Tables 1 and 2) in the genome-scale networks of *Saccharomyces*

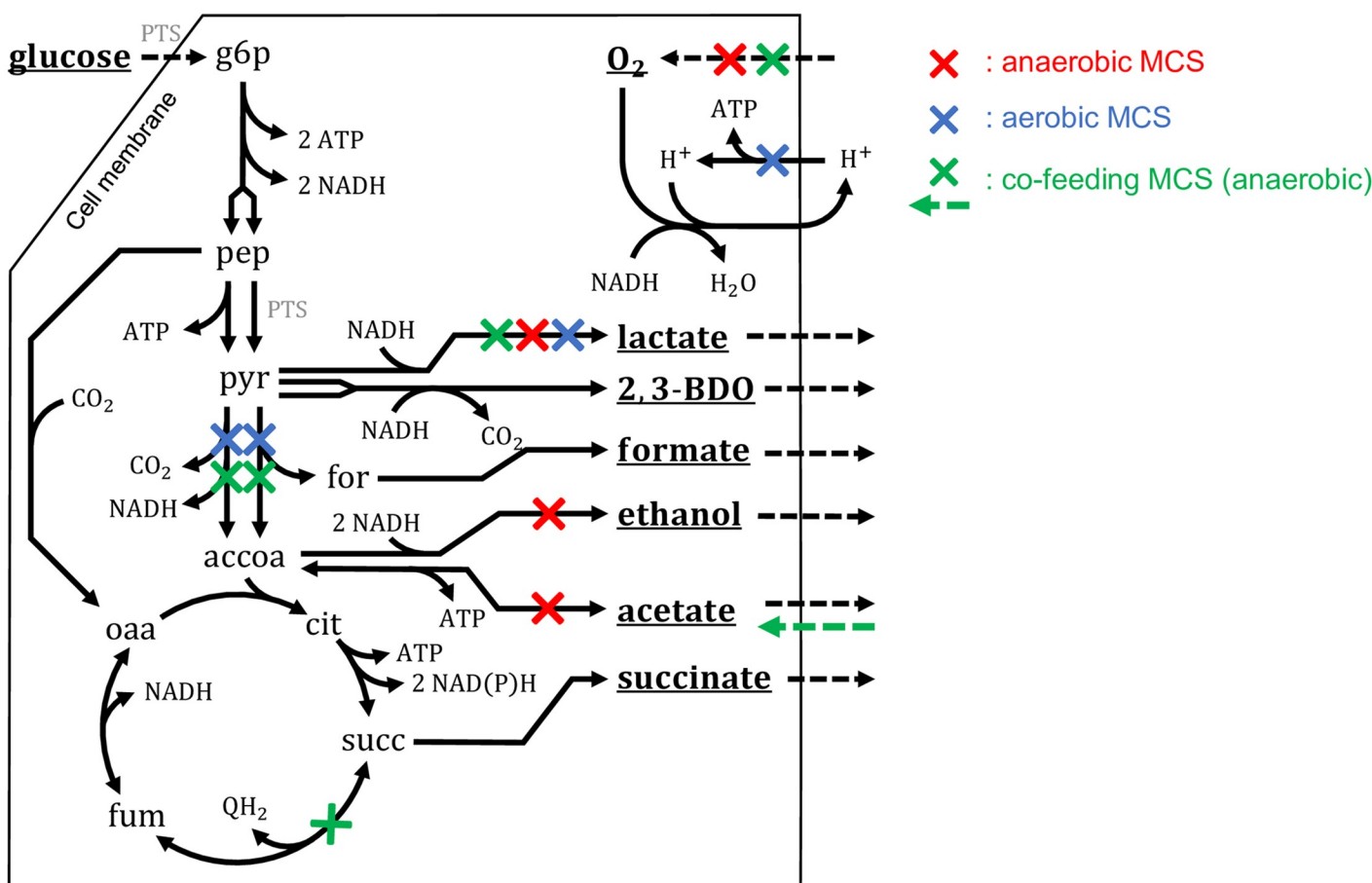

**Fig 6. Main principles of MCS found for the growth-coupled production of 2,3-BDO in E. coli.** Red: anaerobic production (e.g., scenario 1: geneMCS-611 (reaction MCS_043) in S1 Table: -O₂, ΔadhE, ΔadhP, ΔldhA, ΔmgsA, ΔsatP). Blue: aerobic production (scenario 2: geneMCS-449 (reaction MCS_08) in S2 Table: +O₂, ΔaceE, ΔatpA, ΔdeoC, ΔldhA, ΔpflA, ΔpflC, ΔpoxB). Green: anaerobic production with substrate co-feeding (scenario 3: geneMCS-1039 (reaction MCS_095) in S3 Table: -O₂, +acetate, ΔaceE, ΔfrdA, ΔldhA, ΔpflA, ΔpflC, Δpfo).

*cerevisiae* (yeastGEM [61]; 1147 genes and 3989 reactions) and *Pseudomonas putida* (iJN746 [62]; 746 genes and 1054 reactions). We found similar trends in the benchmark computations (see S5 Table). The GPR compression rules are particularly beneficial for the yeast model where we found a speedup factor of 20, while it is 2.2 for the smaller network of *P. putida*. The scripts to reproduce these scenarios are also available under the GitHub link.

## Discussion

We herein introduced several new features and developments generalizing the framework of minimal cut sets and broadening its scope of applications, especially for metabolic network design. We first enhanced the MCS approach by allowing the definition of multiple target (undesired) and multiple protected (desired) regions. These extensions enable now a precise tailoring of metabolic solution spaces of the strain to be designed. We are not aware of any strain design approach that allows such an unlimited flexibility in the description of arbitrary (especially non-convex) spaces of desired and protected metabolic behaviors. Specifying multiple target and desired regions thus represents an entirely new feature and facilitates the treatment of completely new classes of complex design problems. It is not only essential for the computation of co-feeding strategies, where a substrate may also act as potential byproduct,

but also allows one to demand distinct behaviors and responses to different (process) conditions. This could be used for designing production hosts for multi-stage processes, or, as shown in our example calculations, for finding growth-coupled strains designs where the organism can also withstand higher ATP maintenance requirements.

A second extension we introduced is a modified approach for integrating (compressed) gene-protein-reaction (GPR) association in constrained-based models for the computation of intervention strategies. In the context of MCS calculations, so far only few studies attempted to account for GPR associations. One example for such an approach is gMCS [26,27]. In a pre-processing step, this algorithm assesses correspondences between gene knockouts and associated reaction deletions and integrates these relationships in a subsequent gene MCS computation [26]. The authors used their algorithm for computing synthetic lethalities. However, the gMCS algorithms does not support the direct computation of constrained MCS, that maintain a desired behavior as needed, for example, for strain design applications. Moreover, compression techniques have not been proposed for reducing the dimension of the problem. Machado et al. [25] proposed a more general approach for integrating GPR rules in constraint-based metabolic network models. This approach uses pseudo-metabolites and pseudo-reactions to represent genes and their connections with enzymes and reactions in the metabolic model. Among other applications, this integrated representation was also used for the computation of gene MCS [25]. We adopted this approach herein and first introduced a modification that ensures that the original flux space is preserved also in cases where specific flux bounds (different from zero or infinity) are given. Generally, the integration of GPR associations into the metabolic network with this approach largely increases the network size which may limit its application in genome-scale networks. To reduce this overhead, we introduced a set of compression rules that remove redundancies in GPR associations and genes, many of which cannot be addressed by classical network compression alone. The benchmark calculations showed a performance benefit with a factor of 6–8. Importantly, these compression rules for GPR associations can as well be applied in conjunction with other strain optimization methods.

As another major extension, we generalized the MCS framework to now also allow combinations of reaction/gene deletions and additions and to assign arbitrary cost factors to each intervention. We exemplified this feature for finding substrate co-feeding strategies for growth-coupled strain designs, however, other applications include the search for strain designs where, in combination with suitable gene/reaction knockouts, heterologous reactions or pathways are added to the network to achieve optimal production behavior with minimal intervention costs. Clearly, as demonstrated in the application example, offering a larger number of addable reactions may vastly increase the solution space, especially when searching for co-feeding strategies on multiple possible substrates. Hence, the repository of optional insertions should be limited to a manageable number of (preselected) reactions or pathways. Generally, a full enumeration of all MCS in genome-scale models is typically only feasible up to a certain maximal number of interventions (for example, an enumeration of MCS with 9 or more interventions for 2,3-BDO synthesis in iML1515 was not possible due to memory overflow). On the other hand, the calculation of single MCS is often possible also in very large networks.

We note that some of the presented features, in particular substrate co-feeding or reaction additions, have partly been addressed also in previous studies that used bi-level optimization approaches [7,8,33,34,63]. However, neither individual cost factors nor reaction additions and co-feeding strategies have so far been used in the more general framework of MCS. Moreover, co-feeding strategies, where an external metabolite may act as (addable) substrate or

byproduct, requires the definition of multiple target regions and can thus only be handled by the extended MCS approach.

The new algorithmic developments have been integrated in the freely available open source package *CellNetAnalyzer*. The respective functions can identify a single MCS, a certain number of MCS, or enumerate all MCS with a given maximum number of interventions. The extended MCS framework is backwards compatible and can be applied, out of the box, to previous MCS setups as used, for example, by Harder et al. (2016) [16] or von Kamp and Klamt (2017) [47]. The functionality of regulatory MCS [24] can now directly be handled within the new framework: for each reaction, whose flux is considered to be adjustable, several copies of this reactions with different flux bounds are provided as addition candidates. The algorithm will then return MCS that propose the deletion of certain reaction/genes in combination with the addition of the regulated reaction with adequate bounds. A comparative study on computation of synthetic lethals revealed consistency with the gMCS [26,27] method, confirming suitability of our approach also for the identification of synthetic lethals on the genetic level. A potential direction for future research is the combination of our extended MCS framework with the $MCS^2$ algorithm, a recently published variant of the duality-based MCS base algorithm [23], which holds promises to further reduce the MILP size and thus to further speed up MCS computation.

We exemplified the new developments for the MCS framework by computing strain designs for the growth-coupled production of 2,3-butanediol in *E. coli*, *S. cerevisiae and P. putida*. Here, benchmark calculations showed a clear benefit of the new compression rules for GPR associations. The analysis of the computation results for *E. coli* showed that the new features allow the identification of qualitatively new strain designs that could not be generated with existing methods. Previous experimental works favored the use of microaerobic conditions for 2,3-BDO production in *E. coli* [51,52,64]. The employed strain designs are similar to those found in our computed strategies. Alternative fermentation pathways were blocked, e.g. by knockouts of *ldhA*, *adhE*, and *pta*. In addition, the excess of reduction equivalents arising from the unbalanced net stoichiometry of the 2,3-BDO synthesis pathway must be handled by the (limited) formation of another byproduct. This includes $CO_2$ if some amount of oxygen is supplied. Yet, these microaerobic fermentation strategies, which balance the NADH surplus through respiration, are highly sensitive to the ratio of oxygen and substrate uptake rates. Higher oxygen uptake rates lead to an increased $CO_2$ and biomass production and consequently to a decreased 2,3-BDO yield. Low oxygen uptake rates, on the other hand, limit NADH recovery and result in poor cell viability, reduced 2,3-BDO productivity or increased synthesis of reduced by-products like succinate [52]. Our computed strain designs confirm that the simultaneous synthesis of reduced by-products is required to balance the reduction equivalents when operating under fully anaerobic conditions. However, the supply of a potential electron acceptor (e.g. acetate) permits the recovery of reduction equivalents under anaerobic conditions without losing the primary substrate in redox-balancing by-products. We also note that microaerobic strategies for 2,3-BDO synthesis could also be found by our MCS approach by offering several addable oxygen uptake reactions with different maximum uptake rates, which can then be combined with suitable gene deletions to find production strains that can operate under limited oxygen supply.

## Supporting information

**S1 Table. Ranked list of MCS computed in scenario 1 (see Table 2) for growth-coupled 2,3-BDO synthesis in the *E. coli* genome-scale model iML1515.**
(XLSX)

**S2 Table. Ranked list of MCS computed in scenario 2 (see Table 2) for growth-coupled 2,3-BDO synthesis in the *E. coli* genome-scale model iML1515.**
(XLSX)

**S3 Table. Ranked list of MCS computed in scenario 3 (see Table 2) for growth-coupled 2,3-BDO synthesis in the *E. coli* genome-scale model iML1515.**
(XLSX)

**S4 Table. Ranked list of MCS computed in scenario 4 (see Table 2) for growth-coupled 2,3-BDO synthesis in the *E. coli* genome-scale model iML1515.**
(XLSX)

**S5 Table. Results of MCS computations for growth-coupled 2,3-BDO synthesis in the genome-scale models of *S. cerevisiae* and *P. putida*.**
(XLSX)

## Acknowledgments

We thank Elad Noor for providing thermodynamic data for the iML1515 *E. coli* model used for the characterization of the MCS.

## Author Contributions

**Conceptualization:** Philipp Schneider, Steffen Klamt.

**Funding acquisition:** Steffen Klamt.

**Investigation:** Philipp Schneider, Steffen Klamt.

**Methodology:** Philipp Schneider, Axel von Kamp, Steffen Klamt.

**Project administration:** Steffen Klamt.

**Software:** Philipp Schneider, Axel von Kamp, Steffen Klamt.

**Supervision:** Steffen Klamt.

**Validation:** Philipp Schneider, Steffen Klamt.

**Writing – original draft:** Philipp Schneider, Steffen Klamt.

**Writing – review & editing:** Philipp Schneider, Axel von Kamp, Steffen Klamt.

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
