## [Decision Letter · Decision Letter 0]

27 Apr 2020

Dear Dr. Klamt,

Thank you very much for submitting your manuscript "An Extended and Generalized Framework for the Calculation of Metabolic Intervention Strategies Based on Minimal Cut Sets" for consideration at PLOS Computational Biology.

As with all papers reviewed by the journal, your manuscript was reviewed by members of the editorial board and by several independent reviewers. In light of the reviews (below this email), we would like to invite the resubmission of a significantly-revised version that takes into account the reviewers' comments.

We cannot make any decision about publication until we have seen the revised manuscript and your response to the reviewers' comments. Your revised manuscript is also likely to be sent to reviewers for further evaluation.

Sincerely,

Kiran Raosaheb Patil, Ph.D.

Associate Editor

PLOS Computational Biology

Daniel Beard

Deputy Editor

PLOS Computational Biology

Reviewer's Responses to Questions

**Comments to the Authors:**

Reviewer #1: 

The authors extend the already existing CellNetAnalyzer for the computation of MCSs in metabolic networks. The two major additions are the integration of constraints allowing to describe contradicting conditions of the network, e. g. aerobic and anaerobic growth, and compression rules which reduce the size of the network, while including gene rules and therefore not only reactions. <o:p></o:p>

They show the improvements of their method by applying it to the core and genome-scale version of E. Coli. Their computational time is significantly reduced and they were able to compute MCSs in the genome scale network which was not possible before.<o:p></o:p>

 **Overall impression:<o:p></o:p>**

I think the authors made a considerate improvement for the computation of MCSs in genome-scale metabolic networks. However, I think further tasks have to be executed to fully show the general applicability of their method. I am especially concerned about their claim to be able to add new reactions to their metabolic network which they do in fact “by hand”, thus this feature is not provided by their method but is part of the input which is created by the user.<o:p></o:p>

** ****Major:<o:p></o:p>**

- Extension 3: Where does the additional reactions come from?

I do not really see a “proof of concept” for the addition of reactions. Adding new reactions to the network “by hand” is changing the input, thus this is not part of the introduced method or a new “computational” addition. I would rather like to see an automated way, similar to how it was done here https://doi.org/10.1186/1752-0509-6-30 where the KEGG-data base was used.<o:p></o:p>

 - I would like to see a wider application, not only for the two E. Coli models. Maybe the authors can apply their method to all metabolic networks from the BiGG model data base, in order to see what the benchmarks are and the limits (regarding the sizes of the networks) of their method. Right now they claim to introduce a new toolbox, but do not show the overall applicability of it.<o:p></o:p>

- For me the major achievement here is not that their method is faster, but that they were able to compute MCSs in the genome-scale E. Coli which was not possible before. They should emphasize this more. Also by applying it to other genome-scale networks and show where they are able to compute MCSs and the non-decompressed version not.<o:p></o:p>

- Is the computation of MCSs using CellNetAnalyzer the same as the non-decompressed column in table1? If yes, please clarify. If not, the authors should compare their method with CellNetAnalyzer and/or other state-of-the-art MCSs methods.<o:p></o:p>

- page 17: The authors claim that decompressing the MCSs computed in the compressed network may lead to higher number of interventions per MCS. Thus they do not compute MCSs for the original network. Can they elaborate what that means? For example is a MCS of size 1 in the original network always a MCS of size 1 in the compressed network (I think the answer is yes, but not the other way around)? It could happen that a lot of “small” MCSs in the compressed network are actually too large for wet lab applications, thus computing those does not seem to be efficient. Can they compare the size of the compressed MCSs to the size of the MCSs in the original network the compressed MCSs relate to? How do they make sure that all MCSs of desired size are found?<o:p></o:p>

- I would rather like a different structure of the article. I would like to first read the overall results, thus the improvement of their method, and the extensive description of the rules and methods afterwards.<o:p></o:p>

** **

**Minor:<o:p></o:p>**

- Extension 2: What happens if *p_i* is set to 0? Would this imply that reaction *i* is always set to zero (or non-zero). Would you still be guaranteed with a true MCS?<o:p></o:p>

- What does CSOM stand for? Abbreviation is not introduced (p. 3, paragraph 2)<o:p></o:p>

- No introduction of “flux vector” (steady state or not? However, it is defined later) (p. 3, paragraph 2).<o:p></o:p>

- cite https://doi.org/10.1093/bioinformatics/bty1027 on p3-4<o:p></o:p>

- steady state vs. steady**-**state (inconsistent spelling)<o:p></o:p>

- p5: MCS abbreviation was already introduced<o:p></o:p>

- Formula (5): What is *Y*?<o:p></o:p>

- what is *P* and *S* on page 7? Is it O2 up or just some general substrate product? What is *X*? What is *mu*?<o:p></o:p>

 -in Formula (13): 0.25z_{1,p,i} + 0.25 …. <= zi: Do you mean ‘=’ instead of ‘<=’? Otherwise this is redundant or?

I find the equations really hard to read because of how they are presented. Can you put the (in)equality signs all in the center, such that they are aligned?<o:p></o:p>

Do you need z_{1,n,i} = 0 for irreversible reactions? Shouldn’t this be in Ar<=b (as claimed before)?<o:p></o:p>

- What is the difference between constrained MCSs and the first extension? As far as I understood, cMCSs in https://doi.org/10.1093/bioinformatics/btv217 can be also formulated for several desired and undesired phenotypes. However, not for those which interdict each other (e..g aerobic and anaerobic maximum growth). Please clarify.<o:p></o:p>

- I would like to have more explanation on how to derive formula (9) (page 6). And it should be in a better readable format. What size is the identity matrix?<o:p></o:p>

- Extension 3: DOI 10.1007/s10295-014-1576-3 add to references<o:p></o:p>

- page 9: MCS3 and MCS4 are part of the WT too, thus the additional reactions are not necessary here, or?<o:p></o:p>

- page 10, first sentence, typo: additionAL candidates<o:p></o:p>

- page 10: Why can exchange reaction not be deleted? And if so, is this integrated in the MILPs? What if a reaction which is directly coupled to an exchange reaction is deleted?<o:p></o:p>

- page 14: rule 2 and rule 3 can be merged. If a gene is essential for an essential reaction it should be marked as protected. Or am I missing something? Isn’t rule 2 contained in rule 3?<o:p></o:p>

- table 1:<o:p></o:p>

            - EColiCore2, first column “No compression”, rows “# MCS found for compressed network”       and “# MCS found after decompression”: Why 6025 and 6015? Shouldn’t it be the same?<o:p></o:p>

            - what is row “Reactions”? At the top the number of reactions is 502 (core), resp 2715      (genome-scale), but the row contains higher numbers for the non-compressed network. Please     explain. (Including the GPR rules delivers a smaller number too).   Same for the number of             species.

            - I am missing a legend which explains the rows and columns. It is also confusing that the rows   say “compressed network”, even though the columns distinguish between compressed and     decompresses network too. Seems to be redundant and is confusing.<o:p></o:p>

- It is sometimes hard to distinguish between examples and actual constraints for the programs. Can the authors clarify this more? Maybe introducing an “example-environment”.<o:p></o:p>

Reviewer #2: The MCS methodology has been one of the major breakthroughs derived from the field of Systems Biology. The extensions of the MCS approach presented by Professor Klamt and colleagues are really interesting and valuable for different practical applications. In my opinion, this article deserves publication in Plos Computational Biology. However, I have some comments that, in my opinion, could improve the manuscript.

1- For readers not familiar with the MCS approach, the relationship between variables in the dual problem and constraints in the primal could be described in pages 5-6.

2- The objective function for the first extension (multiple targets) could be defined after Eq. (13). By the way, this new approach is really elegant. However, the number of variables is double. In Table 2, you describe the computation time in the case of 2,3-BDO and ATP maintenance. Is it worthy in terms of computation time? Why not resolving 2632*2 (max/min ATP maintenance reaction) LPs to filter those satisfying ATP maintenance?

3- Why the number of MCSs found after decompression in EcoliCore2 is different to the uncompressed case (6015 vs 6025)?

4- I like a lot your approach to gene MCSs. It is clear the effect of GPR compression. However, it would be nice to compare your approach with the one presented in Apaolaza, 2019 in order to calculate synthetic lethals.

**Have all data underlying the figures and results presented in the manuscript been provided?**

Reviewer #1: Yes

Reviewer #2: Yes

PLOS authors have the option to publish the peer review history of their article (what does this mean?). If published, this will include your full peer review and any attached files.

Reviewer #1: No

Reviewer #2: No
---

## [Decision Letter · Decision Letter 1]

30 Jun 2020

Dear Dr. Klamt,

We are pleased to inform you that your manuscript 'An Extended and Generalized Framework for the Calculation of Metabolic Intervention Strategies Based on Minimal Cut Sets' has been provisionally accepted for publication in PLOS Computational Biology.

Best regards,

Kiran Raosaheb Patil, Ph.D.

Associate Editor

PLOS Computational Biology

Daniel Beard

Deputy Editor

PLOS Computational Biology

Reviewer's Responses to Questions

**Comments to the Authors:**

Reviewer #1: The manuscript has significantly improved. I especially want to thank the reviewers for adding two more applications of their method and for answering my questions and clarifying my confusions. Also, the explanation and notation of the formulas is much more clear now.

I have no further comments.

**Have all data underlying the figures and results presented in the manuscript been provided?**

Reviewer #1: Yes

PLOS authors have the option to publish the peer review history of their article (what does this mean?). If published, this will include your full peer review and any attached files.

Reviewer #1: No

---

## [Editor Report · Acceptance letter]

20 Jul 2020

PCOMPBIOL-D-20-00417R1 

An Extended and Generalized Framework for the Calculation of Metabolic Intervention Strategies Based on Minimal Cut Sets

Dear Dr Klamt,

I am pleased to inform you that your manuscript has been formally accepted for publication in PLOS Computational Biology. Your manuscript is now with our production department and you will be notified of the publication date in due course.

With kind regards,

Matt Lyles
